# Chemical Profiling and Antioxidant Activity of Wild and Cultivated Sage (*Salvia officinalis* L.) Essential Oil

**Zoran S. Ilić** [1,*], **Žarko Kevrešan** [2], **Ljubomir Šunić** [1], **Ljiljana Stanojević** [3], **Lidija Milenković** [1], **Jelena Stanojević** [3], **Aleksandra Milenković** [3] and **Dragan Cvetković** [3]

[1] Faculty of Agriculture Priština in Lešak, University of Priština in Kosovska Mitrovica, 38219 Lesak, Serbia; ljubomir.sunic@pr.ac.rs (L.Š.); lidija.milenkovic@pr.ac.rs (L.M.)
[2] Institute of Food Technology, University of Novi Sad, 21000 Novi Sad, Serbia; zarko.kevresan@fins.uns.ac.rs
[3] Faculty of Technology, University of Niš, 16000 Leskovac, Serbia; stanojevic@tf.ni.ac.rs (L.S.); jstanojevic@tf.ni.ac.rs (J.S.); aleksandra.milenkovic@student.ni.ac.rs (A.M.); dragancvetkovic1977@yahoo.com (D.C.)
[*] Correspondence: zorans.ilic@pr.ac.rs; Tel.: +381-63-8014966

**Abstract:** Chemical profiling the sage essential oils (SEOs) from wild and cultivated (shaded or non-shaded) plants has been investigated. The yield of SEOs from wild plants (3.51 mL/100 g) was higher than that from cultivated plants (shaded plants: 3.20 mL/100 g and non-shaded plants: 2.56 mL/100 g). The main components of SEO from wild plants were *cis*-thujone (43.2%), camphor (17.6%), 1,8-cineole (13.8%), veridiflorol (3.8%) and borneol (3.4%). The chemical composition of SEO from cultivated plants included camphor > *cis*-thujone > 1,8-cineole. Net shading lowered the content of toxic *cis*-thujone in sage (23.5%) and is therefore recommended in order to achieve better quality of SEO compared to non-shaded plants (*cis*-thujone 28.3%). The thujone content of SEO from wild plants is much higher (43.2%), and this drastically reduces the quality of EO. Cultivated sage was found to have stronger antioxidant activity (shaded plants 6.16 mg/mL or non-shaded 7.49 ± 0.13 mg/mL) compared to wild sage plants (9.65 mg/mL). The isolated SEOs are good sources of natural antioxidants with potential applications in the food and pharmaceutical industries.

**Keywords:** *Salvia officinalis* L.; wild; cultivated; essential oil; GC/MS analysis; antioxidant activity

## 1. Introduction

Sage (*S. officinalis* L.) is an herbaceous, perennial plant known as common or Dalmatian sage, an evergreen in the form of a bush with woody stems and gray-green leaves. Sage belongs to the Salvia genus and family Lamiaceae, with over 1000 species, grown spontaneously in the Middle East and East Mediterranean areas or cultivated, throughout the world [1]. In southern Europe, it is a widely distributed species with medicinal properties that is increasingly cultivated or grows wild and is collected in, e.g., Greece [2], Turkey [3], Serbia [4,5], Montenegro [6], Croatia [7,8], Albania [9,10], North Macedonia [11] and Bulgaria [12]. The genus Salvia in Serbia includes 14 species [5], while these genus in Turkey contains 99 species, 58 of which are endemic [13]. The scientific name of this perennial plant, *Salvia officinalis,* means savior. In Serbia, it is also known as sage, because it is used to "*incense*" the space and disinfect the air. Sage can be grown wild in nature or cultivated as an attractive plant in the garden and has a long history of culinary and medicinal use [14]. Sage is a popular herb with a warm, spicy flavor that can be used in meat industry [15], for seafood and cheese [16]. In traditional medicine, sage has been used to protect the body against, e.g., oxidative stress, free radical damage, angiogenesis, inflammation, bacterial and viral infection [17]. Sage herbal tea has remedial effects in mild dyspepsia and inflammations in the throat and skin [14]. Ethanolic sage extracts are rich sources of rosmarinic acid and are potent spasmolytic agents [18]. Sage is used in

cosmetics, perfumery and the pharmaceutical industry and has been listed in the European Pharmacopoeia [19].

Unplanned collection of sage from natural habitats, especially in the vicinity of processing facilities and small factories for the extraction of essential oil, could threaten its survival. The production of sage as a raw material for industrial processing and the needs of the pharmaceutical and food industries should be organized on larger areas with adequate agricultural techniques and cultivation methods. Studies on the essential oils (EOs) and antioxidants of wild and cultivated sage are limited [20].

The content of sage essential oil (SEO) varies in individual plant parts. α-pinene and 1,8-cineole are the most present in flower, while in the sage leaf the most common components are camphene, limonene, *cis*-thujone, *trans*-thujone and camphor. EOs isolated from the stems are characterized by linalool as the most abundant component [21]. The chemical composition of sage from central Serbia showed that viridiflorol was the main components, followed by camphor, thujones and verticiol [22] while in North Serbia (Vojvodina province), the predominant SEO components were *cis*-thujone (27.1%) and camphor (19.3%), followed by *trans*-thujone and 1,8-cineole [4]. The chemical composition of the common sage chemotype, has been reported in samples from Turkey [23], Brazil [24], Mexico [25] and Croatia [26] and include α-thujone > camphor > 1,8cineole > β-thujone. These results reveal that SEO composition is highly influenced by genetic and environmental factors [3].

This study aimed to compare the essential oil content, chemical composition and antioxidant activity of wild sage leaves (*Salvia officinalis* L.) collected in Montenegro and cultivated sage leaves grown in shade or non-shade conditions from Serbia.

## 2. Material and Methods

### 2.1. Plant Material and Growing Conditions

The wild-grown sage from background of Herceg Novi (village, Kruševice, with the coordinates of 42°28′30.2″ N 18°31′59.1″ E–42°28′37.4″ N 18°31′50.0″ E) was collected from the vegetation period during2021 and 2022, after the flowering stage, to determine the chemical composition and antioxidant activities of EOs from wild-grown sage. The plant harvest was completed in late August.

The experiment with cultivated sage (from local domestic cultivar in Belgrade Seed-Seeds Company) was carried out in an experimental garden in the village of Moravac in south Serbia (21°42′ E, 43°30′ N, altitude 159 m a.s.l.) between 2020 and 2022. In order to achieve a good yield and sufficient EO, before the crop is formed, it is necessary to incorporate 30–40 kg/ha of nitrogen and 80–100 kg/ha of phosphorus and potassium into the soil with basic autumn fertilization. An amount of 30–40 kg/ha of nitrogen should be incorporated in the spring, before the start of growth, to accelerate sprouting and initial plant growth, and the remaining 25–30 kg/ha of nitrogen should be incorporated immediately after the first mowing. In the following years, a combination of pure NPK nutrients (40–50 kg/ha of each) must be introduced into the soil early in the spring through cultivation, and after the first cutting, fertilizing with 25–30 kg/ha of nitrogen.

Two-month-old seedlings were transplanted at a distance of 70 cm between the rows and 40–50 cm in each row. In the first year, the sage was cut for the first time only in July, and the second time at the beginning of October. An amount of 6–8 t of fresh green mass per hectare was obtained from two harvests, and 1.7–2 t of dry leaves, or 10–12 kg of EO, are obtained from it. Perennial sage was mowed for the first time already in May, and the second time at the end of September. If sage is to be used only for the distillation of essential oil, it is cut in full bloom at the beginning of July.

To protect plants from intensive light radiation and elevated temperatures during the summer months, the sage plants were covered by pearl nets (Polysack, Israel) with a shade index of 50%. Shading can also improve the chemical profiling and antioxidant activity of plants. Combinations of shaded sage and non-shaded sage control plants were replicated three times in a split-plot design.

## 2.2. Clevenger-Hydrodistillation

For hydrodistillation, dried and ground plant material (sage leaves-*Salvia officinalis* L., *folium*) were used (laboratory electric mill Braun Aromatic KSM2) for EO isolation by Clevenger-type hydrodistillation, with hydromodulus (ratio of plant material: water) of 1:10 m/V during 120 min by methods Ilić et al. [27].

## 2.3. Antioxidative Activity
DPPH Assay

The ability of the EO to scavenge free DPPH radicals was determined using the DPPH assay. The details of the method used are given in Stanojevic et al. [28].

## 2.4. Gas Chromatography-Mass Spectrometry (GC/MS) and Gas Chromatography-Flame Ionization Detection (GC/FID) Analysis

The details of the method used are given in Ilić et al. [27] and Milenković et al. [29]. In brief for GC/MS analysis, an Agilent Technologies 7890 B gas chromatograph was used, equipped with capillary column (HP-5MS). The instrument was coupled with a selective 5977 A mass detector. The essential oil samples were dissolved in diethyl ether and injected into the GC. Helium was used as the carrier gas at a constant flow rate of 1 $cm^3$/min. The oven temperature was programmed to increase from 60 °C to 246 °C at a rate of 3 °C/min. The temperatures of the MSD transfer line, ion source, and quadrupole mass analyzer were set at 300 °C, 230 °C and 150 °C, respectively. The ionization voltage was 70 eV, and the mass range analyzed was from $m/z$ 41 to 415. Data processing was performed using MSD Chem Station, Mass Hunter Qualitative Analysis, and AMDIS_32 software. For GC/FID analysis, the experimental conditions were identical to GC/MS. The carrier gas for GC/FID analysis was He with flow at 1 $cm^3$/min, and the flows of make-up gas ($N_2$), fuel gas ($H_2$) and oxidizing gas (Air) were 25, 30 and 400 $cm^3$/min, respectively. The temperature of the flame-ionization detector (FID) was set at 300 °C.

Quantification of the components was achieved using the external standards (β-pinene, 1,8-cineole, citral, limonene, linalool, thymol and γ-terpinene.

## 2.5. Statistical Analysis

The EO yield and antioxidant activity were compared using one-way ANOVA and Duncan's multiple range tests, while principal component analysis (PCA) was used to determine similarity and differences of EO main components and the $EC_{50}$. Statistical analysis was performed using Statistica version 14 (http://tibco.com, accessed on 1 December 2020).

## 3. Results and Discussion
### 3.1. Essential Oil Yield

High variations in SEOs content were characterized primarily by plant origin. However, this variation was influenced by other parameters as well environmental conditions, sage populations, wild or cultivated plants, different cultivation methods (light modification by nets, mineral nutrition, plant age, time and methods of harvest and drying), storage conditions and method of extraction.

The higher EO yields were obtained in the extraction of wild sage (3.51 mL/100 g p.m.) compared to cultivated sage. Method of cultivation, e.g., (with or without shading) significantly affects the EO content of cultivated sage. Thus, in shaded plants (3.20 mL/100), the EO content is significantly higher compared to non-shaded ones (2.56 mL/100 g). The differences between wild and shaded cultivated sage are not statistically significant (Table 1).

The essential oil yield of 25 populations (17 indigenous populations) of Dalmatian sage from different Balkan countries ranged from 0.25 to 3.48%, whereas the northwestern populations tended to have lower essential oil yields compared to the southeastern populations [30].

Genetics and geographical origin are the main parameters influencing SEO quantity and quality together with cultivation methods, plant otnogenesis, day length, light

quantity and quality, temperature oscillations and relative humidity [31]. Different from the previous conclusions, the results of Maric et al. [32] reported that the oil yields of wild sage from central Herzegovina near Mostar varied from 0.29 to 1.07 mL/100 g p.m. depending on the stages of plant development. The SEO yield from Bulgarian sage was 0.93 mL/100 g p.m. [12], while the SEO from Polish sage ranged from 1.16 to 1.35 mL/100 g p.m. [33]. Our studies, similarly confirmed these findings. Sage in our region has a higher EO content compared to sage from other countries (ranging from 1.93 to 3.70 mL/100 g p.m.) [26].

**Table 1.** Sage essential oil (SEO) from the wild and cultivated plants.

| Sample | Essential Oil Yield, mL/100 g p.m. * |
|---|---|
| Sage leaves | |
| Wild plants | 3.51 ± 0.03 [a] |
| Cultivated (non-shaded) plants | 2.56 ± 0.02 [b] |
| Cultivated (shaded) plants | 3.20 ± 0.03 [a] |

* p.m.—plant material. Values followed by the same letter do not significantly differ between the treatments, at $p = 0.05$ according to Duncan's multiple range test.

The dependence of SEO yield on the hydrodistillation time of wild and cultivated sage are shown in Figure 1.

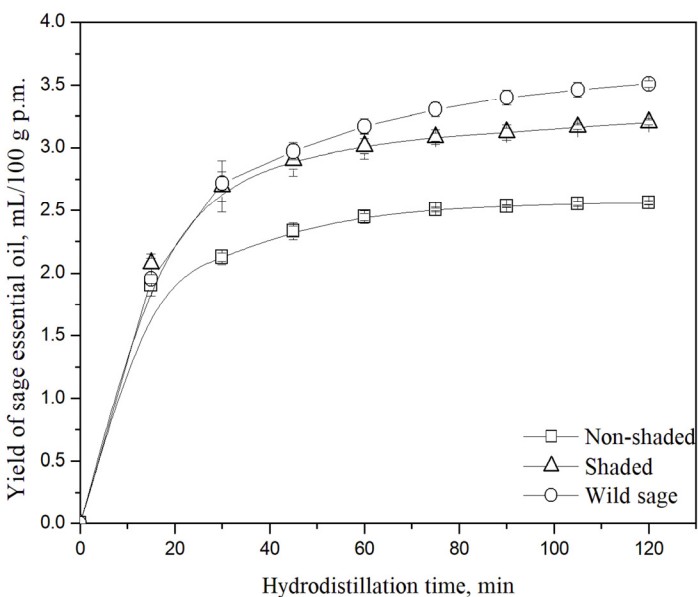

**Figure 1.** SEO Yield of cultivated (shaded and non-shaded) and wild sage depending on the hydrodistillation time.

Medical plants react differently to the shading intensity as well as the color of the shading nets, in improving EO content in many Lamiaceae plants [34], such as sweet basil [35,36] oregano, mint, marjoram, thyme [29,37], lemon balm [37,38] and sage [39].

### 3.2. Essential Oil Composition

Recently, another study identified oxygenated monoterpenes (67.7%) and monoterpene hydrocarbons (19.1%) in the SEOs from Tuscany, Italy [40]. *Cis*-thujone (43.2%), camphor (17.6%) and 1,8-cineole (13.8%) were found to be the most abundant components of SEOs (structures given in Figure 2), followed by veridiflorol (3.8%), borneol (3.4%) and camphene (3%)—Table 2.

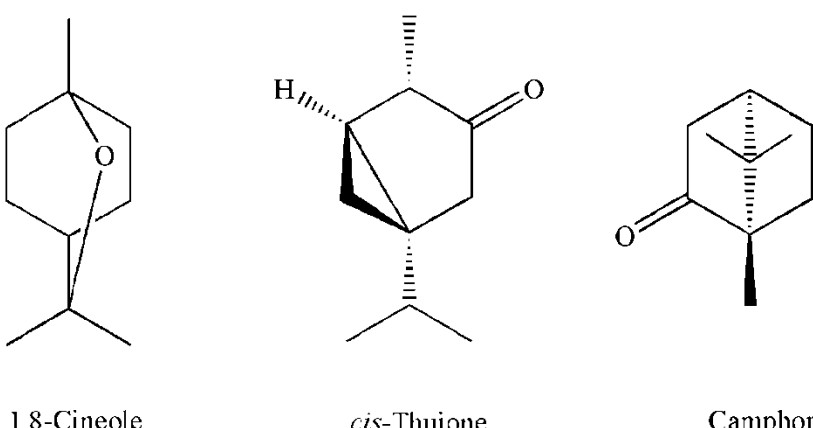

**Figure 2.** Chemical structures of the most abundant components in sage essential oil.

Twenty-nine components were identified in EO from wild sage, mainly oxygen-containing monoterpenes (86.1%), monoterpene hydrocarbons (6.1%), oxygenated sesquiterpenes (4.4%), sesquiterpene hydrocarbons (2.3%) and aromatic compounds (0.8%), constituting 99.8% of the total SEO content(Table 2).

According to the total amount of thujone in SEO Perry et al. [42] grouped sage into three chemotypes (high, 39–44%; middle, 22–28%; low, 9%) and the ratio of α- and β-thujone (α/β 10:1; 1.5:1; 1:10). The sage use in our study belongs to the chemotype with a high content of thujone ranging from 23.5% in shaded plants to 43.2% in wild plants. The ratio of *cis* to *trans*-thujone depends on the origin of the sage plant and the cultivation methods. The *cis* to *trans*-thujone ratio in sage ranges from 1:2.5 in shaded plants to 1:8 in wild plants. The main compounds in SEOs from Serbia were found to be *cis*-thujone, followed by camphor, *trans*-thujone and 1,8-cineole [43]. The main compounds in the essential oil of sage from Serbia were *cis*-thujone, followed by camphor, *trans*-thujone and 1,8-cineole [43].The age of leaves had a significant influence on the SEO composition [44]. In the EO isolated from young leaves, sesquiterpenoid constituents dominated (50.7–57.0%) while EO from development leaves increased the amount of monoterpenoids (55.4–88.4%) with camphor (1.9–32.7%), *cis*-thujone (6.7–28.5%), α-humulene (3.4–33.3%) and viridiflorol (2.9–12.4%) as main components [44]. The major constituents of *S. officinalis* originating from Albania were: camphor (40.2–47.8%), α-thujone (13.7–22.2%), eucalyptol (1,8-cineole) (2.6–6.0%) and camphene (3.9–6.1%), [9]. The chemical composition of SEOs from sage plants grown indifferent region and at different altitudes were characterized high variability. In northern Albania for example, α-thujone chemotype is the most dominant, while in southern Albania the camphor chemotype is the most common [45].

α-thujone, camphor, β-thujene and 1,8-cineole were found to be major constituents of the SEOs analyzed from many European countries [46]. This variability can be attributed to climatic, seasonal, geographical location, geology, physiological and morphological factors, as well as the part of the sage plant and extraction method [24]. The EOs obtained from sage leaves collected in the south-central part of Italy contained α-thujone (7.8–20.1%), camphor (8.4–20.8%), sclareol (5.9–23.1%) and borneol (2.5–16.9%), as the major compounds [47]. In a study by Tundis et al. [48] from Italy high levels of camphor (16.16–18.92%), camphene (6.27–8.08%), 1,8-cineole (8.80–9.86%), β-pinene (3.08–9.14%) and α-thujone (3.08–9.14%) were observed [48]. SEOs from Romania indicated the presence of terpenes such as 1,8-cineole, thujones, borneol, camphor, sabinene, camphene and caryophyllenes as the main component [49]. The EO composition of sage collected in Turkey was β-thujone (34.59%), α-thujone (12.60%) and camphor (10.09%) [13].

Twenty-nine components of EO were identified in cultivated-open field sage, mainly oxygen-containing monoterpenes (75.4%), monoterpene hydrocarbons (10.9%), sesquiterpene hydrocarbons (6.9%) oxygenated sesquiterpenes (0.4%) and diterpenoids (2.1%) (Table 3).

**Table 2.** Chemical composition of essential oil from wild sage.

| No. | $t_{ret.}$, min | Compound | RI$^{exp}$ | RI$^{lit}$ | Method of Identification | Content % |
|---|---|---|---|---|---|---|
| 1 | 6.60 | Santolina triene | 910 | 906 | RI, MS | tr |
| 2 | 6.70 | α-Thujene | 924 | 924 | RI, MS | tr |
| 3 | 6.94 | α-Pinene | 932 | 932 | RI, MS | 1.9 |
| 4 | 7.41 | Camphene | 947 | 946 | RI, MS | 3.0 |
| 5 | 8.28 | β-Pinene | 976 | 974 | RI, MS, Co-I | 0.5 |
| 6 | 8.61 | 1-Octen-3-ol | 977 | 974 | RI, MS | tr |
| 7 | 8.70 | Myrcene | 980 | 988 | RI, MS | 0.8 |
| 8 | 9.24 | α-Phellandrene | 998 | 1002 | RI, MS | tr |
| 9 | 9.66 | α-Terpinene | 1010 | 1014 | RI, MS | tr |
| 10 | 10.04 | p-Cymene | 1020 | 1020 | RI, MS | 0.8 |
| 11 | 10.14 | Limonene | 1022 | 1024 | RI, MS, Co-I | tr |
| 12 | 10.21 | 1,8-Cineole | 1023 | 1026 | RI, MS, Co-I | 13.8 |
| 13 | 11.27 | γ-Terpinene | 1052 | 1054 | RI, MS | tr |
| 14 | 12.45 | Fenchone | 1083 | 1083 | RI, MS | tr |
| 15 | 12.69 | p-Cymenene | 1090 | 1089 | RI, MS | tr |
| 16 | 13.26 | *cis*-Thujone | 1104 | 1101 | RI, MS | 43.2 |
| 17 | 13.73 | *trans*-Thujone | 1115 | 1112 | RI, MS | 5.6 |
| 18 | 14.91 | Camphor | 1144 | 1141 | RI, MS, Co-I | 17.6 |
| 19 | 16.17 | Borneol | 1174 | 1165 | RI, MS, Co-I | 3.4 |
| 20 | 16.51 | Terpinen-4-ol | 1182 | 1174 | RI, MS | 0.8 |
| 21 | 17.27 | α-Terpineol | 1200 | 1196 | RI, MS | tr |
| 22 | 20.75 | Isobornyl acetate | 1282 | 1283 | RI, MS | 1.5 |
| 23 | 21.12 | *trans*-Sabinyl acetate | 1291 | 1289 | RI, MS | 0.2 |
| 24 | 26.47 | (E)-Caryophyllene | 1421 | 1417 | RI, MS | 0.8 |
| 25 | 27.92 | α-Humulene | 1457 | 1452 | RI, MS | 1.5 |
| 26 | 29.03 | Germacrene D | 1485 | 1484 | RI, MS | tr |
| 27 | 33.11 | Caryophyllene oxide | 1591 | 1582 | RI, MS | tr |
| 28 | 33.66 | Veridiflorol | 1602 | 1592 | RI, MS | 3.8 |
| 29 | 34.15 | Humulene epoxide II | 1618 | 1608 | RI, MS | 0.6 |
| | | | | | Total identified | 100.0 |

| Grouped components (%) | |
|---|---|
| Monoterpene hydrocarbons (2–5, 7–11, 13,15) | 7.0 |
| Oxygen-containing monoterpenes (12, 14, 16–23) | 86.2 |
| Sesquiterpene hydrocarbons (24–26) | 2.3 |
| Oxygenated sesquiterpenes (27–29) | 4.5 |
| Others (1, 6) | tr |

$t_{ret.}$: Rete. $t_{ret.}$: Retention time; RI$^{lit}$-Retention indices from the literature (Adams, [41]); RI$^{exp}$: Experimentally determined retention indices using a homologous series of *n*-alkanes (C$_8$–C$_{20}$) on the HP-5MS column. MS: constituent identified by mass-spectra comparison; RI: constituent identified by retention index matching; Co-I: constituent identity confirmed by GC co-injection of an authentic sample; tr = trace amount (<0.05%).

**Table 3.** Chemical profiling of SEO isolated from cultivated plants.

| No. | $t_{ret}$, min | Compound | RI$^{exp}$ | RI$^{lit}$ | Method of Identification | Content % Shaded | Content % Non-Shaded |
|---|---|---|---|---|---|---|---|
| 1 | 4.74 | (Z)-Salvene | 857 | 847 | RI, MS | tr | 0.2 |
| 2 | 4.93 | (E)-Salvene | 868 | 858 | RI, MS | - | tr |
| 3 | 6.41 | Tricyclene | 923 | 921 | RI, MS | tr | 0.1 |
| 4 | 6.51 | α-Thujene | 926 | 924 | RI, MS | 0.2 | 0.2 |
| 5 | 6.73 | α-Pinene | 934 | 932 | RI, MS, Co-I | 2.1 | 4.7 |
| 6 | 7.19 | Camphene | 950 | 946 | RI, MS | 2.9 | 5.4 |
| 7 | 7.90 | Sabinene | 975 | 969 | RI, MS | 0.2 | 0.2 |
| 8 | 8.04 | β-Pinene | 979 | 974 | RI, MS, Co-I | 1.4 | 2.1 |
| 9 | 8.43 | Myrcene | 993 | 988 | RI, MS | 0.9 | 1.1 |
| 10 | 8.94 | α-Phellandrene | 1008 | 1002 | RI, MS | tr | tr |
| 11 | 9.37 | α-Terpinene | 1020 | 1014 | RI, MS | 0.2 | tr |
| 12 | 9.70 | *p*-Cymenene | 1028 | 1020 | RI, MS | 0.1 | 0.2 |
| 13 | 9.82 | Limonene | 1032 | 1024 | RI, MS, Co-I | 1.9 | tr |
| 14 | 9.89 | 1,8-Cineole | 1034 | 1026 | RI, MS, Co-I | 6.1 | 11.0 |
| 15 | 10.09 | (Z)-β-Ocimene | 1039 | 1032 | RI, MS | tr | tr |
| 16 | 10.91 | γ-Terpinene | 1061 | 1054 | RI, MS, Co-I | 0.6 | 0.4 |
| 17 | 11.44 | *cis*-Sabinene hydrate | 1075 | 1065 | RI, MS | tr | tr |
| 18 | 12.06 | Terpinolene | 1092 | 1086 | RI, MS | 0.4 | 0.3 |
| 19 | 12.91 | *cis*-Thujone | 1111 | 1101 | RI, MS | 23.5 | 28.3 |
| 20 | 13.28 | *trans*-Thujone | 1122 | 1112 | RI, MS | 9.0 | 4.4 |
| 21 | 14.51 | Camphor | 1151 | 1141 | RI, MS, Co-I | 31.5 | 30.9 |
| 22 | 15.57 | Borneol | 1175 | 1165 | RI, MS | 3.0 | 2.1 |
| 23 | 15.95 | Terpinen-4-ol | 1184 | 1174 | RI, MS, Co-I | 0.6 | 0.4 |
| 24 | 16.63 | α-Terpineol | 1192 | 1186 | RI, MS | 0.4 | tr |
| 25 | 19.41 | *iso*-3-Thujanol acetate | 1267 | 1267 | RI, MS | tr | - |
| 26 | 20.28 | Isobornyl acetate | 1287 | 1283 | RI, MS | 1.3 | 0.9 |
| 27 | 20.61 | *trans*-Sabinyl acetate* | 1289 | 1295 | RI, MS | tr | - |
| 28 | 21.98 | Myrtenyl acetate* | 1328 | 1324 | RI, MS | tr | - |
| 29 | 25.92 | (E)-Caryophyllene | 1423 | 1417 | RI, MS | 2.5 | 1.3 |
| 30 | 27.32 | α-Humulene | 1458 | 1452 | RI, MS | 4.4 | 2.0 |
| 31 | 32.92 | Khusimone | 1601 | 1604 | RI, MS | 4.5 | 2.1 |
| 32 | 33.46 | Humulene epoxide II | 1616 | 1619 | RI, MS | 0.4 | - |
| 33 | 48.54 | Manool | 2063 | 2056 | RI, MS | 2.1 | 1.4 |
| | | | | | Total identified (%) | 100.0 | 100.0 |
| | Grouped components (%) | | | | | | |
| | Monoterpene hydrocarbons (2–12, 14, 15, 17) | | | | | 10.9 | 14.7 |
| | Oxygen-containing monoterpenes (13, 16, 18–27) | | | | | 75.2 | 78.0 |
| | Sesquiterpene hydrocarbons (28, 29, 30) | | | | | 6.9 | 3.3 |
| | Oxygen-containing sesquiterpenes (32) | | | | | 0.4 | - |
| | Diterpenoids (33) | | | | | 2.1 | 1.4 |
| | Others (1, 31) | | | | | 4.5 | 2.3 |

$t_{ret.}$: Retention time; RI$^{lit}$-Retention indices from the literature (Adams, [42]); RI$^{exp}$: Experimentally determined retention indices using a homologous series of *n*-alkanes (C$_8$–C$_{20}$) on the HP-5MS column. MS: constituent identified by mass-spectra comparison; RI: constituent identified by retention index matching; Co-I: constituent identity confirmed by GC co-injection of an authentic sample; tr = trace amount (<0.05%), n.i.—not identified.

Thirty-two EO components from shaded sage plants were identified, mainly oxygen-containing monoterpenes (78%), monoterpene hydrocarbons (14.7%), sesquiterpene hydrocarbons (3.3%) and diterpenoids (1.4%), representing 99.7% of the total SEO content (Table 3).

Chemical profiling of cultivated sage included camphor > α-thujone > 1,8-cineole. Environmental conditions with light modification using shade nets affected constituents and content of SEO. Thus, shaded plants contained less *cis*-thujone (23.5%) than non-shaded plants (28.3%) Table 3. Shading affects the lower content of toxic *cis*-thujone in sage, and it is recommended to apply net shading in order to achieve a better quality of EO. Comparing the thujone content of cultivated sage (23.5–28.3%) with that of wild sage (43.2%), revealed higher thujone levels in wild sage, which drastically reduces the quality of the essential oil obtained. Results from this study indicated that the different light conditions and temperature created by pearl shade nets influenced the SEO composition. Different external (light, temperature) and internal factors (origin, genotypes, ontogenesis)affect the accumulation of thujone.

Similar to our findings, SEOs from North Macedonia were composed of camphor (13.15–25.91%), α-thujone (19.25–26.33%), β-thujone (2.03–5.28%), 1,8-cineole (6.51–13.60%), α-humulene (2.89–7.99%) and viridiflorol (4.27–7.99%) [11]. Bulgarian SEOs consisted of α-thujone (26.68%), (*E*)-β-caryophyllene (7.47%), 1,8-α-cineole (7.19%), α-humulene (6.11%), β-pinene (5.44%), β-thujone (5.35%) and camphor (4.84%) [12]. SEOs from Brazil, however, contain α-thujone (40.90%), camphor (26.12%), α-pinene (5.85%) and β-thujone (5.62%) [50].

Plant bioactive compounds can be affected by environment. In optimal sage growing conditions, camphor is mostly accumulated, while in dry years, due to heat and water stress, thujone accumulates more [51].

The high content of toxic cis-thujones seems to be prevalent in SEOs from sage plant cultivated in Estonia [46].Thujone is a natural oxygen-containing monoterpene present in two forms: cis-thujone and trans-thujone, in variable amounts in sage. Ratio of cis-thujone and trans-thujone are the main indicators for determining the different chemotypes of sage [4].

Presence of toxic thujones, has been regulated by proposition and law with limited concentration [52]. Thujone in chemically pure form is not allowed to be added to food [19]. A low level of thujone in sage leaf can be used in the preparation of food products such as meat dishes, sage-flavoured sausages, cheese additives, sweets, salad dressings or alcoholic beverages at acceptable levels with a daily intake of up to 5.0 mg of thujone for a maximum of 2 weeks [53].

Previous research on natural sage populations in Serbia revealed that α-thujone (28.2%) was the main EO component, while β-thujone occupied only 5.1% [54]. In our research we have found that sage we can be classified into two groups based on thujone content; (1)wild Montenegrin sage with high (48.8%) total thujone content and (2) cultivated sage from Serbia with medium thujone content (32.6%) resulting in different rations between cis—and trans-thujones (cis/trans 8:1 in wild and 6:1 to 2.5:1 in cultivated sage).

Wild sage from Montenegro showed high variability in EO composition, especially the thujone content [55–57] *cis*-thujone (16.98–40.35%), camphor (12.75–35.37%), 1,8-cineol (6.40–12.06%), *trans*-thujone (1.5–10.35%), were the main SEO constituents in Montenegrin sage [57]. Sage collected from different regions in southern Italy contained EOs with dominant presence of camphor (16.16–18.92%), 1,8-cineole (8.80–9.86%), β-pinene (3.08–9.14%), camphene (6.27–8.08%) and α-thujone (1.17–9.26%) [48].

The major chemical components of sage from Nepal were camphor (65.18%), camphene (9.73%) and 1,8-cineole (4.72%) [58]. Interestingly, thujone was not identified in sage, and plants from these regions are completely safe. The reason for the absence of thujone in Nepalese sage could be the high altitude and the specificity of the Himalayan area. We have similarly found that, α-thujone content in SEO from Turkey decreased with shading level [59]. Several groups have addressed the influence of light modification

on EO metabolism in different plant species [40,60,61]. The EO content of marigold [62], chamomile [63] and yarrow [64] decreased with shading level, whereas sage [39] and oregano [29] were improved by limited light intensity.

Limited results have been published on the chemical profiling of SEO extracted from shading plants [59].

According to our results, the chemical composition of SEO varied significantly with cultivation conditions, and we found that the cultivation of sage under protected conditions was beneficial for major SEO compounds.

### 3.3. Antioxidative Activity

The antioxidant DPPH radical scavenging activity of SEOs of wild and cultivated *S. officinalis* plants were evaluated under different cultivation conditions (Figure 3).

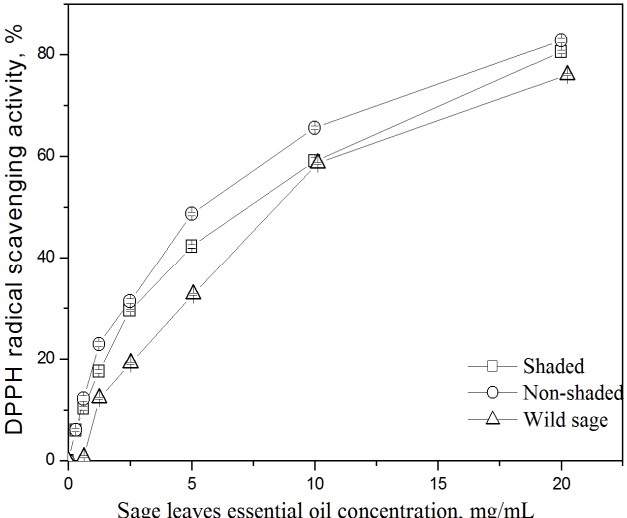

**Figure 3.** Antioxidant activity of SEOs from wild and cultivated (shaded and nonshaded) plants.

The concentrations of SEOs required to reduce the initial radical concentration by 50% ($EC_{50}$) were determined. A lower $EC_{50}$ value indicates a higher antioxidant activity.

As shown in Table 4, as well as from Figure 3 comparing the data, it appears that the antioxidant activity of EOs isolated from the different sage plants (wild and cultivated) during the 20 min incubation decreased in the following order: cultivated shaded sage (6.16 mg/mL) > cultivated non-shaded sage (7.49 mg/mL) > wild sage (9.65 mg/mL). BHT was used as the reference compound ($EC_{50} = 0.021$ mg/mL) [65].

**Table 4.** $EC_{50}$ values of SEOs from the wild and cultivated plants.

| Essential Oil | $EC_{50}$, mg/mL, 20 min Incubation |
|---|---|
| Wild sage | $9.65 \pm 0.01$ [c] |
| Cultivated sage (non-shaded plants) | $7.49 \pm 0.13$ [b] |
| Cultivated sage (shaded plants) | $6.16 \pm 0.06$ [a] |

Values followed by the same letter do not significantly differ between the treatments, at $p = 0.05$ according to Duncan's multiple range test.

The SEOs of a sage plant from Tunisia, with an $EC_{50}$ value of 8.31 mg/L [66] were comparable with the Turkish SEOs previously analyzed by Bouaziz et al. [67] who found an $EC_{50}$ value of 7.70 mg/L. The antioxidant activity of different species within the genus Salvia differs, so it ranges from 2.49 to 7.71 µg/mL according to the determined $EC_{50}$ values [68]. Consistent with our studies with sage, similar findings have been reported for oregano, where shaded oregano plants were found to have higher antioxidant capacity ($EC_{50}$ 7.91 mg/mL) than nonshaded oregano plants ($EC_{50}$ 8.59 mg/mL) [37]. EOs seem to

play a key role in the antioxidant activity of sage. Many studies have unraveled their specific role by using DPPH, TEAEC, ABTS, FRAP or β-carotene bleaching assays. Components found in EOs mainly playing this antioxidant activity were α-thujone, β-thujone, camphor, linalool, 1,8-cineole and others [1].

With the goal to establish a correlation between $EC_{50}$ and the percentage of individual EO components a PCA analysis was performed (Figures 4 and 5). According to the bi-plot in Figure 4, there is no significant differentiation between EOs of sages according to dominant Factor 1. However, separation could be seen with Factor 2.

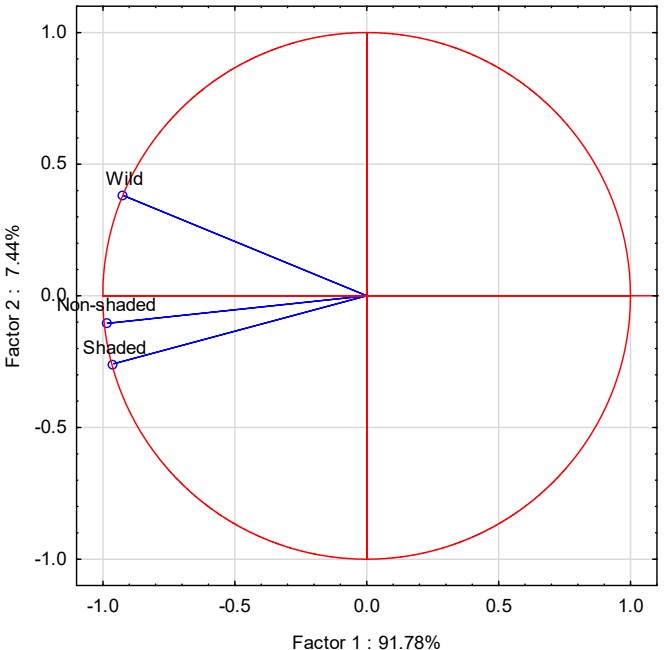

**Figure 4.** Bi-plot of principal component analysis of antioxidant potential of essential oils from wild and cultivated sage.

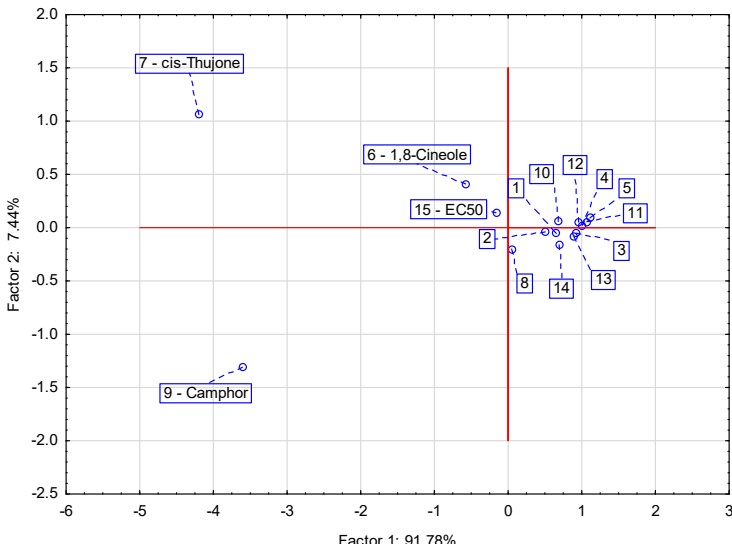

**Figure 5.** Bi-plot of principal component analysis of EO compounds and $EC_{50}$ from wild and cultivated sage. 1; α-Pinene, 2; Camphene, 3; β-Pinene, 4; Myrcene, 5; p-Cymene, 6; 1,8-Cineole, 7; cis-Thujone, 8; trans-Thujone, 9; Camphor, 10; Borneol, 11; Terpinen-4-ol, 12; Isobornyl acetate, 13; (E) -Caryophyllene, 14; α-Humulene, 15; $EC_{50}$.

A bi-plot of individual components is presented in Figure 5. According to data presented in Tables 2 and 3, a separation of components by Factor 1 could be correlated by values, and separation by Factor 2 could be attributed to EO correlation, e.g., compounds below 0 lines are increasing as the $EC_{50}$ values decreases.

In conclusion, cultural practices and methods, in combination with environmental conditions, appear to affect the antioxidant activities of sage plants. The comparison of cultivation conditions (open field or net-house) gives an understanding of how cultural practices in combination with environmental factors (different origin regions) affect the variation in antioxidant content of sage plants. Based on this research, production conditions can be optimized in order to obtain plants richer in EO content with higher antioxidant level.

## 4. Conclusions

Due to global warming, there is a need for urgent diversification, collection, study and preservation of the most resistant populations and ecotypes of *Salvia officinalis* before they disappear and are lost. The principle of studying and preserving the gene pool of sage is also reflected in the fact that only wild sage contains the component veridiflorol in SEO, and is not detected in cultivated sage. Finally, this investigation provides new cultivation methods which affect the content and quality of SEOs. To produce higher content and a better composition of SEO, it is necessary to combine new cultivation methods that optimize the concentration of particular volatile compounds. Camphor was the major compound (31.5%) found in cultivate sage from Serbia under shaded conditions and in open field conditions (30.9%) which is significantly higher than wild sage (17.6%). *Cis*-thujone was the other major compound in the EOs of cultivated sage (23.5–28.3%) but it is the main component in wild sage (43.2%). Based on these results, the accumulation of harmful thujone can be reduced by the cultivation technique, i.e., by applying shading. The antioxidant activity of cultivated sage was higher under shading conditions (6.16 mg/mL) than in open field conditions (7.49 mg/mL) and significantly different than that of wild sage (9.65 mg/mL).Under different growth conditions and cultivation methods, some changes can occur in the EO composition, which may influence the quality of the target compound and therapeutic value of sage.

**Author Contributions:** Z.S.I. and L.S. were heads of the research group, planned the research, analyzed it, and wrote the manuscript; L.M. and L.Š. conducted the experiment in the field; Ž.K. Software, visualization, statistical analysis and J.S., A.M. and D.C. performed analyses on physical properties and chemical composition in the laboratory. All authors have read and agreed to the published version of the manuscript.

**Funding:** This research received external funding from a program for financing scientific research work, with grant numbers 451-03-68/2022-14/200133 and 451-03-68/2022-14/200189 was financially supported by the Ministry of Education Science and Technological Development of the Republic of Serbia.

**Institutional Review Board Statement:** Not applicable.

**Data Availability Statement:** All the data are available in the manuscript file.

**Conflicts of Interest:** There are no conflict of interest.

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
