# Peer review of "Chemical Profiling and Antioxidant Activity of Wild and Cultivated Sage (Salvia officinalis L.) Essential Oil"

_horticulturae, doi:10.3390/horticulturae9060624_

Round 1
Reviewer 1 Report
Dear Authors,
The manuscript with title "Chemical profiling and antioxidant activity of wild and cultivated sage (Salvia officinalis L.) essential oil is very interesting and deals with chemical composition and antioxidant activity of wild, cultivated shade and cultivated non-shaded plants. This manuscript can have applicative input in agronomy for cultivation of sage as well as the impact of climatic condition on chemical composition and antioxidant activity of sage. However, I would like to suggest better discussion in few points:
1. Which components are the most meritorious for antioxidant activity of essential oils of sage, is it camphor itself or synergetic effect between few compounds found in sage?
2. The authors should compare the antioxidant activity of wild and cultivated sage by different assays (ABTS, FRAP etc) in the published literature and discuss the sensitivity of different radicals (assays) on the chemical structure of the major chemical compounds found in sage.
3. It can be interesting to emphasize the antimicrobial and antiradical potential of the major compounds found in wild and cultivated sage.
I suggest acceptance of the manuscript after major revision.
Author Response
The manuscript with title "Chemical profiling and antioxidant activity of wild and cultivated sage (Salvia officinalis L.) essential oil is very interesting and deals with chemical composition and antioxidant activity of wild, cultivated shade and cultivated non-shaded plants. This manuscript can have applicative input in agronomy for cultivation of sage as well as the impact of climatic condition on chemical composition and antioxidant activity of sage. However, I would like to suggest better discussion in few points:
- Which components are the most meritorious for antioxidant activity of essential oils of sage, is it camphor itself or synergetic effect between few compounds found in sage?
Camphor is the most meritorious for antioxidant activity (positive correlation between level of camphor and antioxidant activity), cultivated sage shaded plants !!!
- The authors should compare the antioxidant activity of wild and cultivated sage by different assays (ABTS, FRAP etc) in the published literature and discuss the sensitivity of different radicals (assays) on the chemical structure of the major chemical compounds found in sage.
EOs seem to play a key role in the antioxidant activity of sage. Many studies have unraveled their specific role by using DPPH, TEAEC, ABTS, FRAP, or β-carotene bleaching assays. Components found in EOs mainly playing this antioxidant activity were α-thujone, β-thujone, camphor, linalool, 1,8-cineole, and others (Poulious et al., 2020).
Poulios, E., Giaginis, C., Georgios K. Vasios. 2020. Current state of the art on the antioxidant activity of sage (Salvia spp.) and its bioactive components. Planta Med. 86: 224-238.
- It can be interesting to emphasize the antimicrobial and antiradical potential of the major compounds found in wild and cultivated sage.
We plane to prolong and extend our investigation and to include in experiments antimicrobial properties of SEO very soon.
I suggest acceptance of the manuscript after major revision
We try to improved scientific quality of MS include major revision before being considered for publication
We would like to thank the Editor and Reviewer for their comments and suggestions to improve our manuscript. We hope that in current form our manuscript will meet high standards for publishing in your journal Horticulturae MDPI
Sincerely Yours
Prof dr Zoran Ilic

Reviewer 2 Report
The article determined the chemical profile and antioxidant activity of the essential oil of sage plants. The research topic is relevant, the introduction, materials and methods are described in sufficient detail.
I have a few questions and comments:
Line 72. "The harvest of plants was completed in late August." However, line 70 indicates that the plants were grown until the end of September.
Line 73-86. The start and end dates of flowering are not specified.
Specify the cultivar or genetic origin of cultivated sage.
Not indicated hydrodistillation were dried or cut plants.
Line 202-205. The retention indices and the percentage of undefined components should be supplemented so that the sum in the table would be 100%. Otherwise, the data looks incomplete.
Line 219. "Shading affects the lower content of toxic thujone in sage..." Table 3, line 3 does not show this.
Figure 1 and 2. It is not clear why the figure is divided, if possible, parts a and b should be combined.
Line 312. How were the components separated to determine this? Add a link to the methodology that was used.
Line 327. "(different origin regions) affect the variation in antioxidant content of 327 sage plants" Add a link to the study.
Line 335. "the component veridiflorol in its essential oil, which was not detected in the cultivated one." It should be noted that it was not found according to the applied research method, this does not mean that this component does not exist.
Line 336. "Finally, this investigation provides new cultivation methods with a combination of environmental conditions that affect the quantity and quality of sage plant 337 essential oil." Environmental conditions were not investigated in this study.
Line 345. The standard substance against which concentration the antioxidant activity was determined is not indicated; it should be indicated which substance was used.
Author Response
I have a few questions and comments:
Line 72. "The harvest of plants was completed in late August." However, line 70 indicates that the plants were grown until the end of September.
It’s a mistake.. we have corrected it …
Line 73-86. The start and end dates of flowering are not specified.
Flowering start with third week of June and finished at first week of July.
Specify the cultivar or genetic origin of cultivated sage.
The experiment with cultivated sage (local domestic cultivar from Belgrade Seed-Seeds Company)
Not indicated hydrodistillation were dried or cut plants.
For hydrodistillation, dried and ground plant material (sage leaves-Salvia officinalis L., folium) was used (laboratory electric mill Braun Aromatic KSM2
Line 202-205. The retention indices and the percentage of undefined components should be supplemented so that the sum in the table would be 100%. Otherwise, the data looks incomplete.
The retention index and the percentage of an unidentified component are supplemented in Table 3 (non-shaded sample, component eluted at 33.48 min with RI=1617). The data given in Table 2 and Table 3 (shaded sample) were checked and corrected.
Line 219. "Shading affects the lower content of toxic thujone in sage..." Table 3, line 3 does not show this.
word thujone should be replaced with cis-thujone
Figure 1 and 2. It is not clear why the figure is divided, if possible, parts a and b should be combined.
After you recommendation, we constructed new Figure….. with combined parts a i b
Line 312. How were the components separated to determine this? Add a link to the methodology that was used.
Separation of individual components of essential oil was done by Gas chromatography-mass spectrometry (GC/MS) and gas chromatography-flame ionization detec-101 tion (GC/FID) analysis. Since same method of separation was already described (Ilić et al., 2022, [28] and Milenković et al., 2021.) we did not think that further description is needed. However after you question following text was added in material and method section, chapter 2.4. Gas chromatography-mass spectrometry (GC/MS) and gas chromatography-flame ionization detection (GC/FID) analysis:
In brief for GC/MS analysis, an Agilent Technologies 7890 B gas chromatograph was used, equipped with capillary column (HP-5MS). The instrument was coupled with a selective 5977 A mass detector. The essential oil samples were dissolved in diethyl ether and injected into the GC. Helium was used as the carrier gas at a constant flow rate of 1 cm3/min. The oven temperature was programmed to increase from 60 °C to 246 °C at a rate of 3 °C/min. The temperatures of the MSD transfer line, ion source, and quadrupole mass analyzer were set at 300 °C, 230 °C, and 150 °C, respectively. The ionization voltage was 70 eV, and the mass range analyzed was from m/z 41 to 415. Data processing was performed using MSD ChemStation, MassHunter Qualitative Analysis, and AMDIS_32 software. For GC/FID analysis, the experimental conditions were identical to GC/MS. The carrier gas for GC/FID analysis was He with flow at 1 cm3/min, and the flows of make-up gas (N2), fuel gas (H2), and oxidizing gas (Air) were 25, 30, and 400 cm3/min, respectively. The temperature of the flame-ionization detector (FID) was set at 300 °C.
Quantification of the components was achieved using the external standards (β-pinene, 1,8-cineole, citral, limonene, linalool, thymol, and γ-terpinene.
Separation of individual components shown on Figure 4 was done according to PCA analysis performed with Statistica software, version 14 (TIBCO Software Inc. (2020). Data Science Workbench. http://tibco.com) as described in chapter 2.5. Statistical analysis
Line 327. "(different origin regions) affect the variation in antioxidant content of 327 sage plants" Add a link to the study.
Wild sage was originated from Montenegro and cultivated sage originates from central Serbia.
Line 335. "the component veridiflorol in its essential oil, which was not detected in the cultivated one." It should be noted that it was not found according to the applied research method, this does not mean that this component does not exist.
We are accept your suggestion and include in text
Line 336. "Finally, this investigation provides new cultivation methods with a combination of environmental conditions that affect the quantity and quality of sage plant 337 essential oil." Environmental conditions were not investigated in this study.
We are exclude part of these sentence …..with a combination of environmental conditions that
Line 345. The standard substance against which concentration the antioxidant activity was determined is not indicated; it should be indicated which substance was used
BHT was used as the reference compound (EC50 = 0.021 mg/mL) (Stanojević et al., 2016).
Ref: Stanojević, L., Stanković, M., Cvetković, D., Danilović, B., Stanojević, J. Dill (Anethum graveolens L.) seeds essential oil as a potential natural antioxidant and antimicrobial agent, Biologica Nyssana, 7 (1) 2016, 31-39.
We would like to thank the Editor and Reviewer for their comments and suggestions to improve our manuscript. We hope that in current form our manuscript will meet high standards for publishing in your journal Horticulturae MDPI
Sincerely Yours
Prof dr Zoran Ilic

Reviewer 3 Report
It is very interesting research. Here some corrections and suggestions for the authors.
Page 1, line 32
Correct punctuation. See attach file.
Page 1, line 34
Change ”...Greece [2],Turkey...” to “...Greece [2], Turkey...”. Add space after comma.
Page 1, line 35
In lower case; change "Flora" to "flora".
Page 1, line 37
A period must be added after "[13]".
Page 2, lines 74-88.
It must be written in the past tense.
Page 3, line 96 and 103
Change "Ilić et al., 2022 [28]." to "Ilić et al. [28]."
Page 3, line 103
For “Milenković et al., 2021.” Add the reference number that corresponds to it.
Page 3, line 105
It must be written in the past tense.
Page 3, line 107
In "EC50", the number "50" must be in subscript.
Page 3, line 108
Change “(TIBCO Software Inc. (2020)” to “(TIBCO Software Inc. (2020))”.
Page 5, line 153
Figure 2 does not correspond to structures of the compounds.
Page 6, line 158
The citation must be corrected according to the instructions for authors. “(Adams, 2007)”.
Page 8, line 208
The citation must be corrected according to the instructions for authors. “(Adams, 2007)”.
Page 9, line 246
Correct citation “(EMA/HMPC 2008b).”
Page 9, line 253
Change “(cis/trans 8:1 in wild and 6:1 to 2.5:1 in cultivated sage.” To “(cis/trans 8:1 in wild and 6:1 to 2.5:1 in cultivated sage).”. The parenthesis must be closed. Add ")".
Page 10, line 279
Change "2 a" to "2a".
Page 10, line 287
In “EC50”, the number "50" must be in subscript.
Page 10, line 295
Should it be EC50? Otherwise, establish the differences or similarities between IC50 and EC50.
See attached file for details.

Author Response
It is very interesting research.
Here some corrections and suggestions for the authors.
Page 1, line 32 Correct punctuation.
Page 1, line 34 Change ”...Greece [2],Turkey...” to “...Greece [2], Turkey...”. Add space after comma.
Page 1, line 35 In lower case; change "Flora" to "flora". Page 1, line 37 A period must be added after "[13]".
We make correction…
Page 2, lines 74-88. It must be written in the past tense.
Yes we changes
Page 3, line 96 and 103 Change "Ilić et al., 2022 [28]." to "Ilić et al. [28]."
yes we do it [28].
Page 3, line 103 For “Milenković et al., 2021.” Add the reference number that corresponds to it.
[29]
2 Page 3, line 105 It must be written in the past tense.
We accept you suggestion
Page 3, line 107 In "EC50", the number "50" must be in subscript.
Yes will be in subscript
Page 3, line 108 Change “(TIBCO Software Inc. (2020)” to “(TIBCO Software Inc. (2020))”.
Yes we add ))
Page 5, line 153 Figure 2 does not correspond to structures of the compounds.
We add new Figures 2. with structures
Page 6, line 158 The citation must be corrected according to the instructions for authors. “(Adams, 2007)”.
[42]
Page 8, line 208 The citation must be corrected according to the instructions for authors. “(Adams, 2007)”.
[42]
Page 9, line 246 Correct citation “(EMA/HMPC 2008b).”
[55]
Page 9, line 253 Change “(cis/trans 8:1 in wild and 6:1 to 2.5:1 in cultivated sage.” To “(cis/trans 8:1 in wild and 6:1 to 2.5:1 in cultivated sage).”. The parenthesis must be closed. Add ")".
Yes we add parenthesis at end.
Page 10, line 279 Change "2 a" to "2a".
We changes this
Page 10, line 287 In “EC50”, the number "50" must be in subscript.
We do it
Page 10, line 295 Should it be EC50? Otherwise, establish the differences or similarities between IC50 and EC50. See attached file for details.
EC50 was correct !!!!!

Round 2
Reviewer 1 Report
Dear Authors,
The revised version of manuscript with title "Chemical profiling and antioxidant activity of wild and cultivated sage (Salvia officinalis L.) essential oil" is significantly improved and all questions and suggestions are corrected and answered. I suggest acceptance of the manuscript in this revised form.